# EXPRESSIVE MODELING IS INSUFFICIENT FOR OFFLINE RL: A TRACTABLE INFERENCE PERSPECTIVE

## ABSTRACT

A popular paradigm for offline Reinforcement Learning (RL) tasks is to first fit the offline trajectories to a sequence model, and then prompt the model for actions that lead to high expected return. While a common consensus is that more expressive sequence models imply better performance, this paper highlights that *tractability*, the ability to exactly and efficiently answer various probabilistic queries, plays an equally important role. Specifically, due to the fundamental stochasticity from the offline data-collection policies and the environment dynamics, highly non-trivial conditional/constrained generation is required to elicit rewarding actions. While it is still possible to approximate such queries, we observe that such crude estimates significantly undermine the benefits brought by expressive sequence models. To overcome this problem, this paper proposes **Trifle** (**Tr**actable **I**nference for Of**fl**ine RL), which leverages modern Tractable Probabilistic Models (TPMs) to bridge the gap between good sequence models and high expected returns at evaluation time. Empirically, Trifle achieves the most state-of-the-art scores in 9 Gym-MuJoCo benchmarks against strong baselines. Further, owing to its tractability, Trifle significantly outperforms prior approaches in stochastic environments and safe RL tasks (e.g., with action constraints) with minimum algorithmic modifications.

## 1 INTRODUCTION

Recent advancements in deep generative models have opened up the possibility of solving offline Reinforcement Learning (RL) (Levine et al., 2020) tasks with sequence modeling techniques (termed RvS approaches). Specifically, we first fit the trajectories provided in an offline dataset to a sequence model. During evaluation, the model is tasked to sample actions with high expected returns given the current state. Leveraging modern deep generative models such as GPTs (Brown et al., 2020) and diffusion models (Ho et al., 2020), RvS algorithms have significantly boosted the performance on various discrete/continuous control problems (Ajay et al., 2022; Chen et al., 2021).

Despite its appealing simplicity, it is still unclear whether expressive modeling alone guarantees good performance of RvS algorithms, and if so, on what types of environments. Perhaps surprisingly, this paper discovers that many common failures of RvS algorithms are not caused by modeling problems. Instead, while useful information is encoded in the model during training, the model is unable to elicit such knowledge during evaluation. Specifically, this issue is reflected in two aspects: (i) *inability to accurately estimate the expected return* of a state and a corresponding action sequence to be executed given near-perfect learned transition dynamics and reward functions; (ii) even when accurate return estimates exist in the offline dataset and are learned by the model, it could still *fail to sample rewarding actions* during evaluation.[1] At the heart of such inferior evaluation-time performance is the fact that highly non-trivial conditional/constrained generation is required to stimulate high-return actions (Paster et al., 2022; Brandfonbrener et al., 2022). Therefore, other than expressiveness, the ability to efficiently and exactly answer various queries (e.g., computing the expected cumulative rewards), termed *tractability*, plays an equally important role in RvS approaches.

Having observed that the lack of tractability is an essential cause of the underperformance of RvS algorithms, this paper studies *whether we can gain practical benefits from using Tractable Probabilistic Models (TPMs) (Poon & Domingos, 2011; Choi et al., 2020; Kisa et al., 2014), which*

---

[1]Both observations are supported by empirical evidence as illustrated in Section 3.

*by design support exact and efficient computation of certain queries?* We answer the question in its affirmative by showing that we can leverage a class of TPMs that support computing arbitrary marginal probabilities to significantly mitigate the inference-time suboptimality of RvS approaches. The proposed algorithm **Trifle** (**Tr**actable **I**nference for O**ffl**in**e** RL) has three main contributions:

*Emphasizing the important role of tractable models in offline RL.* This is the first paper that demonstrates the possibility of using TPMs on complex offline RL tasks. The superior empirical performance of Trifle suggests that expressive modeling is not the only aspect that determines the performance of RvS algorithms, and motivates the development of better inference-aware RvS approaches.

*Competitive empirical performance.* Compared against strong offline RL baselines (including RvS, imitation learning, and offline temporal-difference algorithms), Trifle achieves the most state-of-the-art scores on 9 Gym-MuJoCo benchmarks (Fu et al., 2020).

*Generalizability to stochastic environments and safe-RL tasks.* Trifle can be extended to tackle stochastic environments as well as safe RL tasks with minimum algorithmic modifications. Specifically, we evaluate Trifle in a stochastic Taxi environment and action-space-constrained MuJoCo environments, and demonstrate superior performance against all baselines.

## 2 PRELIMINARIES

**Offline Reinforcement Learning** In Reinforcement Learning (RL), an agent interacts with an unknown environment at discrete time steps to maximize its cumulative reward. The environment is defined by a Markov Decision Process (MDP) $\langle \mathcal{S}, \mathcal{A}, \mathcal{R}, \mathcal{P}, d_0 \rangle$, where $\mathcal{S}$ is the state space, $\mathcal{A}$ is the action space, $\mathcal{R} : \mathcal{S} \times \mathcal{A} \to \mathbb{R}$ is the reward function, $\mathcal{P} : \mathcal{S} \times \mathcal{A} \to \mathcal{S}$ is the transition dynamics, and $d_0$ is the initial state distribution. Our goal is to learn a policy $\pi(a|s)$ that maximizes the expected return $\mathbb{E}[\sum_{t=0}^{T} \gamma^t r_t]$, where $\gamma \in (0, 1]$ is a discount factor and $T$ is the maximum number of steps.

Offline RL (Levine et al., 2020) aims to solve RL problems where we cannot freely interact with the environment. Instead, we receive a dataset of trajectories collected using unknown policies. An effective learning paradigm for offline RL is to treat it as a sequence modeling problem (termed RL via Sequence Modeling or RvS methods) (Janner et al., 2021; Chen et al., 2021; Emmons et al., 2021). Specifically, we first learn a sequence model on the dataset, and then sample actions conditioned on past states and high future returns. Since the models typically do not encode the entire trajectory, an estimated value or return-to-go (RTG) (i.e., the Monte Carlo estimate of the sum of future rewards) is also included for every state-action pair, allowing the model to estimate the return at any time step.

**Tractable Probabilistic Models** Tractable Probabilistic Models (TPMs) are generative models that are designed to efficiently and exactly answer a wide range of probabilistic queries (Poon & Domingos, 2011; Choi et al., 2020; Rahman et al., 2014). One example class of TPMs is Hidden Markov Models (HMMs) (Rabiner & Juang, 1986), which support linear time (w.r.t. model size and input size) computation of marginal probabilities and more. Recent advancements have extensively pushed forward the expressiveness of modern TPMs (Liu et al., 2022; 2023; Correia et al., 2023), leading to competitive likelihoods on natural image and text datasets compared to even strong Variational Autoencoder (Vahdat & Kautz, 2020) and Diffusion model (Kingma et al., 2021) baselines. This paper leverages such advances and explores the benefits brought by TPMs in offline RL tasks.

**Probabilistic Circuits** Probabilistic circuits (PCs) (Choi et al., 2020) are a general and unified computational framework for tractable probabilistic modeling, i.e., a wide class of TPMs can be represented as a PC. Similar to neural networks, PCs are also computation graphs containing millions of computation units, where constraints are imposed on their structures to enable efficient computation of various probabilistic queries. Generally, computing probabilistic queries involves executing specific feedforward/backward algorithms over the PC-defined computation graph. More details about TPM and PC are shown in Appx. B and C.

## 3 TRACTABILITY MATTERS IN OFFLINE RL

Practical RvS approaches operate in two main phases – training and evaluation. In the training phase, a sequence model is adopted to learn a joint distribution over trajectories of length $T$: $\{(s_t, a_t, r_t, \mathrm{RTG}_t)\}_{t=0}^{T}.$[2] During evaluation, at every time step $t$, the model is tasked to discover an

---

[2]To minimize computation cost, we only model truncated trajectories of length $K$ ($K < T$) in practice.

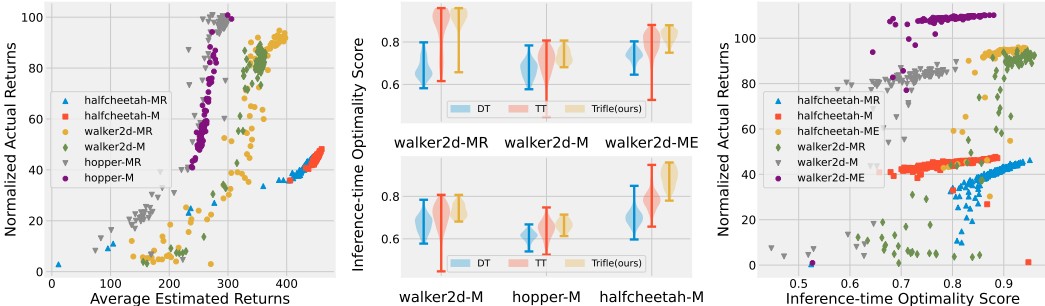

Figure 1: RvS approaches suffer from inference-time suboptimality. **Left:** There exists a strong positive correlation between the average estimated returns by Trajectory Transformers (TT) and the normalized actual returns in 6 Gym-MuJoCo environments (MR, M, and ME are short for medium-replay, medium, and medium-expert, respectively), which suggests that the sequence model can distinguish rewarding actions from bad ones. **Middle:** Despite being able to recognize high-return actions, both TT and DT (Chen et al., 2021) fail to consistently sample such action, leading to bad inference-time optimality; our method Trifle consistently improves the inference-time optimality score. **Right:** We substantiate the relationship between low inference-time optimality scores and unfavorable environmental outcomes by showing a strong positive correlation between them.

action sequence $a_{t:T} := \{a_\tau\}_{\tau=t}^T$ (or just $a_t$) that has high expected return as well as high probability in the prior policy $p(a_{t:T}|s_t)$, which prevents it from generating out-of-distribution actions:

$$p(a_{t:T}|s_t, \mathbb{E}[V_t] \geq v) := \frac{1}{Z} \cdot \begin{cases} p(a_{t:T}|s_t) & \text{if } \mathbb{E}_{V_t \sim p(\cdot|s_t, a_t)}[V_t] \geq v, \\ 0 & \text{otherwise,} \end{cases} \quad (1)$$

where $Z$ is a normalizing constant, $V_t$ is an estimate of the value at time step $t$, and $v$ is a pre-defined scalar whose value is chosen to encourage high-return policies. Depending on the problem, $V_t$ could be the labeled RTG from the dataset (e.g., $\text{RTG}_t$) or the sum of future rewards capped with a value estimate (e.g., $\sum_{\tau=t}^{T-1} r_\tau + \text{RTG}_T$) (Emmons et al., 2021; Janner et al., 2021).

The above definition naturally reveals two key challenges in RvS approaches: (i) *training-time optimality* (i.e., "expressivity"): how well can we fit the offline trajectories, and (ii) *inference-time optimality*: whether actions can be unbiasedly and efficiently sampled from Equation (1). While extensive breakthroughs have been achieved to improve the training-time optimality (Ajay et al., 2022; Chen et al., 2021; Janner et al., 2021), it remains unclear whether the non-trivial constrained generation task of Equation (1) hinders inference-time optimality. In the following, we present two general scenarios where existing RvS approaches underperform as a result of suboptimal inference-time performance. We attribute such failures to the fact that these models are limited to answering certain query classes (e.g., autoregressive models can only compute next token probabilities), and explore the potential of *tractable* probabilistic models for offline RL tasks in the following sections.

**Scenario #1** We first consider the case where the labeled RTG belongs to a (near-)optimal policy. In this case, Equation (1) can be simplified to $p(a_t|s_t, \mathbb{E}[V_t] \geq v)$ (choose $V_t := \text{RTG}_t$) since one-step optimality implies multi-step optimality. In practice, although the RTGs are suboptimal, the predicted values often match well with the actual returns achieved by the agent. Take Trajectory Transformer (TT) (Janner et al., 2021) as an example, Figure 1 (left) demonstrates a strong positive correlation between its predicted returns (x-axis) and the actual cumulative rewards (y-axis) on six MuJoCo (Todorov et al., 2012) benchmarks, suggesting that the model has learned the "goodness" of most actions. In such cases, the performance of RvS algorithms depends mainly on their inference-time optimality, i.e., whether they can efficiently sample actions with high *predicted* returns. Specifically, let $a_t$ be the action taken by a RvS algorithm at state $s_t$, and $R_t := \mathbb{E}[\text{RTG}_t]$ is the corresponding estimated expected value. We define a proxy of inference-time optimality as the quantile value of $R_t$ in the estimated state-conditioned value distribution $p(V_t|s_t)$.[3] The higher the quantile value, the more frequent the RvS algorithm samples actions with high estimated returns.

We evaluate the inference-time optimality of Decision Transformers (DT) (Chen et al., 2021) and Trajectory Transformers (TT) (Janner et al., 2021), two widely used RvS algorithms, on various

---

[3]Due to the large action space, it is impractical to compute $p(V_t|s_t) := \sum_{a_t} p(V_t|s_t, a_t) \cdot p(a_t|s_t)$. Instead, in the following illustrative experiments, we train an additional GPT model $p(V_t|s_t)$ using the offline dataset.

environments and offline datasets from the Gym-MuJoCo benchmark suite (Fu et al., 2020). Perhaps surprisingly, as shown in Figure 1 (middle), the inference-time optimality is averaged (only) around 0.7 (the maximum possible value is 1.0) for most settings. And these runs with low inference-time optimality scores receive low environment returns (Fig. 1 (right)).

**Scenario #2** Achieving inference-time optimality becomes even harder when the labeled RTGs are suboptimal (e.g., they come from a random policy). In this case, even estimating the expected future return of an action sequence becomes highly intractable, especially when the transition dynamics of the environment are stochastic. Specifically, to evaluate a state-action pair $(s_t, a_t)$, since $\mathrm{RTG}_t$ is uninformative, we need to resort to the multi-step estimate $V_t^{\mathrm{m}} := \sum_{\tau=t}^{t'-1} r_\tau + \mathrm{RTG}_{t'}$ ($t' > t$), where the actions $a_{t:t'}$ are jointly chosen to maximize the expected return. Take autoregressive models as an example. Since the variables are arranged following the sequential order $\ldots, s_t, a_t, r_t, \mathrm{RTG}_t, s_{t+1}, \ldots$, we need to explicitly sample $s_{t+1:t'}$ before proceed to compute the rewards and the RTG in $V_t^{\mathrm{m}}$. When the transition dynamics are stochastic, estimating $\mathbb{E}[V_t^{\mathrm{m}}]$ could suffer from high variance as the stochasticity from the intermediate states accumulates over time.

As we shall illustrate in Section 6.2, compared to environments with near-deterministic transition dynamics (e.g., Fig. 1 (left)), estimating the expected returns in stochastic environments using intractable sequence models is hard, and Trifle can significantly mitigate this problem with its ability to marginalize out intermediate states and compute $\mathbb{E}[V_t^{\mathrm{m}}]$ in closed-form.

## 4 EXPLOITING TRACTABLE MODELS

The previous section demonstrates that apart from modeling, inference-time suboptimality is another key factor that causes the underperformance of RvS approaches. Given such observations, a natural follow-up question is *whether/how more tractable models can improve the evaluation-time performance in offline RL tasks?* While there are different types of tractabilities (i.e., the ability to compute different types of queries), this paper focuses on studying the additional benefit of *exactly* computing *arbitrary* marginal/condition probabilities. This strikes a proper balance between learning and inference as we can train such a tractable yet expressive model thanks to recent developments in the TPM community (Liu et al., 2022; Correia et al., 2023). Note that in addition to proposing a competitive RvS algorithm, we aim to highlight the necessity and benefit of using more tractable models for offline RL tasks, and encourage future developments on both inference-aware RvS methods and better TPMs. As a direct response to the two failing scenarios identified in Section 3, in the following, we first demonstrate how tractability could help even when the labeled RTGs are (near-)optimal (Sec. 4.1). We then move on to the more general case where we need to use multi-step return estimates to account for biases in the labeled RTGs (Sec. 4.2).

### 4.1 FROM THE SINGLE-STEP CASE...

Consider the case where the RTGs are optimal. Recall from Section 3 that our goal is to sample actions from $p(a_t|s_t, \mathbb{E}[V_t] \geq v)$ ($V_t := \mathrm{RTG}_t$). Prior works use two typical ways to approximately sample from this distribution. The first approach directly trains a model to generate return-conditioned actions: $p(a_t|s_t, \mathrm{RTG}_t)$ (Chen et al., 2021). However, since the RTG given a state-action pair is stochastic,[4] sampling from this RTG-conditioned policy could result in actions with a small probability of getting a high return, but with a low expected return (Paster et al., 2022; Brandfonbrener et al., 2022).

An alternative approach leverages the ability of sequence models to accurately estimate the expected return (i.e., $\mathbb{E}[\mathrm{RTG}_t]$) of state-action pairs (Janner et al., 2021). Specifically, we first sample from a prior distribution $p(a_t|s_t)$, and then reject actions with low expected returns. Such rejection sampling-based methods typically work well when the action space is small (in which we can enumerate all actions) or the dataset contains many high-rewarding trajectories (in which the rejection rate is low). However, the action could be multi-dimensional and the dataset typically contains many more low-return trajectories in practice, rendering the inference-time optimality score low (cf. Fig. 1).

Having examined the pros and cons of existing approaches, we are left with the question of whether a tractable model can improve sampled actions (in this single-step case). We answer it with a mixture of positive and negative results: while computing $p(a_t|s_t, \mathbb{E}[V_t] \geq v)$ is NP-hard even when $p(a_t, V_t|s_t)$

---

[4]This is true unless (i) the policy that generates the offline dataset is deterministic, (ii) the transition dynamics is deterministic, and (iii) the reward function is deterministic.

follows a simple Naive Bayes distribution, we can design an approximation algorithm that samples high-return actions with high probability in practice. We start with the negative result.

**Theorem 1.** *Let $a_t := \{a_t^i\}_{i=1}^k$ be a set of $k$ boolean variables and $V_t$ be a categorical variables with two categories $0$ and $1$. For some $s_t$, assume the joint distribution over $a_t$ and $V_t$ conditioned on $s_t$ follows a Naive Bayes distribution: $p(a_t, V_t|s_t) := p(V_t|s_t) \cdot \prod_{i=1}^k p(a_t^i|V_t, s_t)$, where $a_t^i$ denotes the $i$th variable of $a_t$. Computing $p(a_t|s_t, \mathbb{E}[V_t] \geq v)$ is NP-hard.*

The proof is given in Appx. A. While it seems hard to directly draw samples from $p(a_t|s_t, \mathbb{E}[V_t] \geq v)$, we propose to improve the aforementioned rejection sampling-based method by adding a correction term to the original proposal distribution $p(a_t|s_t)$ to reduce the rejection rate. Specifically, the prior is often represented by an autoregressive model such as GPT: $p_{\text{GPT}}(a_t|s_t) := \prod_{i=1}^k p_{\text{GPT}}(a_t^i|s_t, a_t^{<i})$, where $k$ is the number of action variables and $a_t^i$ is the $i$th variable of $a_t$. We sample every dimension of $a_t$ autoregressively following:

$$\forall i \in \{1, \ldots, k\} \quad \tilde{p}(a_t^i|s_t, a_t^{<i}; v) := \frac{1}{Z} \cdot p_{\text{GPT}}(a_t^i|s_t, a_t^{<i}) \cdot p_{\text{TPM}}(V_t \geq v|s_t, a_t^{\leq i}), \quad (2)$$

where $Z$ is a normalizing constant and $p_{\text{TPM}}(V_t \geq v|s_t, a_t^{\leq i})$ is a correction term that leverages the ability of the TPM to compute the distribution of $V_t$ given incomplete actions. Note that while Equation (2) is mathematically identical to $p(a_t|s_t, V_t \geq v)$, this formulation gives us the flexibility to use the prior policy (i.e., $p_{\text{GPT}}(a_t^i|s_t, a_t^{<i})$) represented by more expressive autoregressive generative models. Further, as shown in Figure 1 (middle), compared to using $p(a_t|s_t)$ (as done by TT), the inference-time optimality scores increase significantly when using the distribution specified by Equation (2) (as done by Trifle) across various Gym-MuJoCo benchmarks.

## 4.2 ...TO THE MULTI-STEP CASE

Recall that when the labeled RTGs are suboptimal, our goal is to sample from $p(a_{t:t'}|s_t, \mathbb{E}[V_t^{\text{m}}] \geq v)$ ($t' > t$), where $V_t^{\text{m}} := \sum_{\tau=t}^{t'-1} r_\tau + \text{RTG}_{t'}$ is the multi-step value estimate. However, as elaborated in the second scenario in Section 3, it is hard even to evaluate the expected return of an action sequence due to the inability to marginalize out intermediate states $s_{t+1:t'}$. Empowered by TPMs, we can readily solve this problem thanks to the linearity of the expectation operator:

$$\mathbb{E}\big[V_t^{\text{m}}\big] = \sum_{\tau=t}^{t'-1} \mathbb{E}_{r_\tau \sim p(\cdot|s_t, a_{t:t'})}\big[r_\tau\big] + \mathbb{E}_{\text{RTG}_{t'} \sim p(\cdot|s_t, a_{t:t'})}\big[\text{RTG}_{t'}\big].$$

We are now left with the same problem discussed in the single-step case — how to sample actions with high expected returns (i.e., $\mathbb{E}[V_t^{\text{m}}]$). Following Equation (2), we sample potentially rewarding actions by conditioning the action sequence on $V_t^{\text{m}} \geq v$:

$$\tilde{p}(a_{t:t'}|s_t; v) := \prod_{\tau=t}^{t'} \tilde{p}(a_\tau|s_t, a_{<\tau}; v), \text{ where } \tilde{p}(a_\tau|s_t, a_{<\tau}; v) \propto p(a_\tau|s_t, a_{<\tau}) \cdot p(V_t^{\text{m}} \geq v|s_t, a_{\leq \tau}),$$

$a_{<\tau}$ and $a_{\leq \tau}$ represent $a_{t:\tau-1}$ and $a_{t:\tau}$, respectively.[5] In practice, while we compute $p(V_t \geq v|s_t, a_{\leq \tau})$ using the TPM, $p(a_\tau|s_t, a_{<\tau}) = \mathbb{E}_{s_{t+1:\tau}}[p(a_\tau|s_{\leq \tau}, a_{<\tau})]$ can either be computed exactly with the TPM or approximated (via Monte Carlo estimation over $s_{t+1:\tau}$) using an autoregressive neural network. In summary, we approximate samples from $p(a_{t:t'}|s_t, \mathbb{E}[V_t] \geq v)$ by first sampling from $\tilde{p}(a_{t:t'}|s_t; v)$, and then rejecting samples whose (predicted) expected return is smaller than $v$.

## 5 PRACTICAL IMPLEMENTATION WITH TPMS

The previous section has demonstrated how to efficiently sample from the expected-value-conditioned policy (Eq. 1). Based on this sampling algorithm, this section further introduces the proposed algorithm **Trifle** (**Tr**actable **I**nference for O**ffl**ine RL). The high-level idea of Trifle is to obtain good action (sequence) candidates from $p(a_t|s_t, \mathbb{E}[V] \geq v)$, and then use beam search to further single out the most rewarding action. Intuitively, by the definition in Equation (1), the candidates are both

---

[5]We approximate $p(V_t^{\text{m}} \geq v|s_t, a_{\leq \tau})$ by assuming that the variables $\{r_t, \ldots, r_{t'-1}, \text{RTG}_{t'}\}$ are independent. Specifically, we first compute $\{p(r_\tau|s_t, a_{\leq \tau})\}_{\tau=t}^{t'-1}$ and $p(\text{RTG}_{t'}|s_t, a_{\leq \tau})$, and then sum up the random variables assuming that they are independent. This holds strictly for environments with deterministic transition dynamics and remains a decent approximation for stochastic environments.

Table 1: Normalized Scores on the standard Gym-MuJoCo benchmarks. The results of Trifle are averaged over 12 random seeds, and mean as well as standard deviations are reported. Results of the baselines are acquired from their respective papers. "# Wins of Trifle" denotes the number of environments out of 9 where Trifle outperforms each baseline. Bold indicates the best result.

| Dataset | Environment | RvS | | | | | IL | | | Offline-TD | | |
|---|---|---|---|---|---|---|---|---|---|---|---|---|
| | | **Trifle** | BR-RCRL | DD | TT | DT | BC | 10%BC | TD3+BC | IQL | CQL | BEAR |
| Med-Expert | HalfCheetah | $95.1_{\pm0.3}$ | **95.2** | 90.6 | 95.0 | 86.8 | 55.2 | 92.9 | 90.7 | 86.7 | 91.6 | 53.4 |
| Med-Expert | Hopper | $113.0_{\pm0.4}$ | 112.9 | 111.8 | 110.0 | 107.6 | 52.5 | 110.9 | 98.0 | 91.5 | 105.4 | 96.3 |
| Med-Expert | Walker2d | $109.3_{\pm0.1}$ | **111.0** | 108.8 | 101.9 | 108.1 | 107.5 | 109.0 | 110.1 | 109.6 | 108.8 | 40.1 |
| Medium | HalfCheetah | $49.5_{\pm0.2}$ | 48.6 | 49.1 | 46.9 | 42.6 | 42.6 | 42.5 | 48.3 | 47.4 | 44.0 | 41.7 |
| Medium | Hopper | $67.1_{\pm4.3}$ | 78.0 | **79.3** | 61.1 | 67.6 | 52.9 | 56.9 | 59.3 | 66.3 | 58.5 | 52.1 |
| Medium | Walker2d | $83.1_{\pm0.8}$ | 82.3 | 82.5 | 79.0 | 74.0 | 75.3 | 75.0 | **83.7** | 78.3 | 72.5 | 59.1 |
| Med-Replay | HalfCheetah | $45.0_{\pm0.3}$ | 42.3 | 39.3 | 41.9 | 36.6 | 36.6 | 40.6 | 44.6 | 44.2 | **45.5** | 38.6 |
| Med-Replay | Hopper | $97.8_{\pm0.3}$ | 98.3 | **100.0** | 91.5 | 82.7 | 18.1 | 75.9 | 60.9 | 94.7 | 95.0 | 33.7 |
| Med-Replay | Walker2d | $88.3_{\pm3.8}$ | 80.6 | 75.0 | 82.6 | 66.6 | 26.0 | 62.5 | 81.8 | 73.9 | 77.2 | 19.2 |
| # Wins of Trifle | Reference | | 5 | 7 | 9 | 8 | 9 | 9 | 8 | 8 | 8 | 9 |

rewarding and have relatively high likelihoods in the offline dataset, which ensures the actions are within the offline data distribution and prevents overconfident estimates during beam search.

Beam search maintains a set of $N$ (incomplete) sequences each starting as an empty sequence. For ease of presentation, we assume the current time step is 0. At every time step $t$, beam search replicates each of the $N$ actions sequences into $\lambda \in \mathbb{Z}^+$ copies and appends an action $a_t$ to every sequence. Specifically, for every partial action sequence $a_{<t}$, we sample an action following $p(a_t|s_0, a_{<t}, \mathbb{E}[V_t] \geq v)$, where $V_t$ can be either the single-step or the multi-step estimate depending on the task. Now that we have $\lambda \cdot N$ trajectories in total, the next step is to evaluate their expected return, which can be computed exactly using the TPM (see Sec. 4.2). The $N$-best action sequences are kept and proceed to the next time step. After repeating this procedure for $H$ time steps, we return the best action sequence. The first action in the sequence is used to interact with the environment.

Another design choice is the threshold value $v$. While it is common to use a fixed high return throughout the episode, we follow Ding et al. (2023) and use an adaptive threshold. Specifically, at state $s_t$, we choose $v$ to be the $\epsilon$-quantile value of $p(V_t|s_t)$, which is computed using the TPM.

**Details of the adopted TPM** This paper uses Probabilistic Circuits (PCs) (Choi et al., 2020) as an example TPM to demonstrate the effectiveness of Trifle. PCs can compute any marginal or conditional probability in time linear with respect to its size. We use a state-of-the-art PC structure called HCLT (Liu & Van den Broeck, 2021) and optimize its parameters using the latent variable distillation technique (Liu et al., 2022). Additional training/inference details are provided in Appx. C.

## 6 EXPERIMENTS

This section takes gradual steps to study whether Trifle can mitigate the inference-time suboptimality problem in different settings. First, in the case where the labeled RTGs are good performance indicators (i.e., the single-step case), we examine whether Trifle can consistently sample more rewarding actions (Sec. 6.1). Next, we further challenge Trifle in highly stochastic environments, where existing RvS algorithms fail catastrophically due to the failure to account for the environmental randomness (Sec. 6.2). Finally, we demonstrate that Trifle can be directly applied to safe RL tasks (with action constraints) by effectively conditioning on the constraints being satisfied (Sec. 6.3). Collectively, this section highlights the potential of TPMs on offline RL tasks.

### 6.1 COMPARISON WITH THE STATE OF THE ART

As demonstrated in Section 3 and Figure 1, although the labeled RTGs in the Gym-MuJoCo (Fu et al., 2020) benchmarks are accurate enough to reflect the actual environmental return, existing RvS algorithms fail to effectively sample such actions due to their large and multi-dimensional action space. Figure 1 (middle) has demonstrated that Trifle achieves better inference-time optimality. This section further examines whether higher inference-time optimality scores lead to better performance.

**Environment setup** The Gym-MuJoCo benchmark suite collects trajectories in 3 locomotion environments (HalfCheetah, Hopper, Walker2D) and constructs 3 datasets (Medium-Expert, Medium,

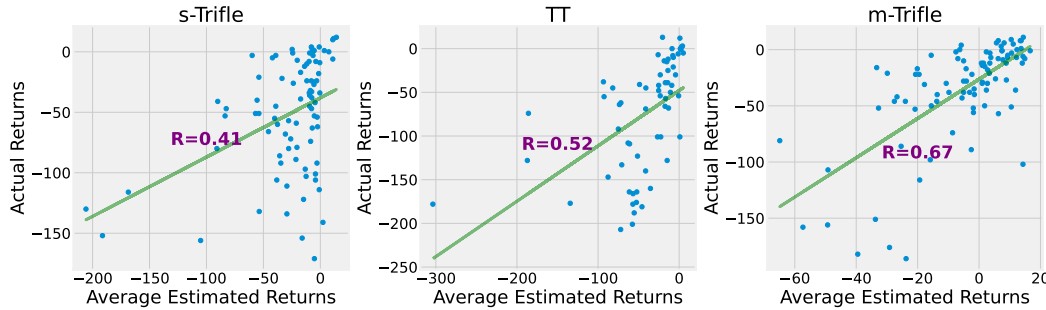

Figure 2: Correlation between average estimated returns and true environmental returns for s-Trifle (w/ single-step value estimates), TT, and m-Trifle (w/ multi-step value estimates) in the stochastic Taxi domain. $R$ denotes the correlation coefficient. The results demonstrate that (i) multi-step value estimates (TT and m-Trifle) are better than single-step estimates (s-Trifle), and (ii) exactly computed multi-step estimates (m-Trifle) are better than approximated ones (TT) in stochastic environments.

Medium-Replay) for every environment, which results in $3 \times 3 = 9$ tasks. For every environment, the main difference between the datasets is the quality of its trajectories. Specifically, the dataset "Medium" records 1 million steps collected from a Soft Actor-Critic (SAC) (Haarnoja et al., 2018) agent. The "Medium-Replay" dataset contains all samples in the replay buffer recorded during the training process of the SAC agent. The "Medium-Expert" dataset mixes 1 million steps of expert demonstrations and 1 million suboptimal steps generated by a partially trained SAC policy or a random policy. The results are normalized to ensure that the well-trained SAC model has a 100 score and the random policy has a 0 score.

**Baselines** We compare Trifle against three main classes of offline RL methods: (i) RvS approaches including Trajectory Transformer (TT) (Janner et al., 2021), Decision Transformer (DT) (Chen et al., 2021), Decision Diffuser (DD) (Ajay et al., 2022), and Bayesian Repramaterized RCRL (BR-RCRL) (Ding et al., 2023); (ii) Offline TD learning methods (Offline-TD) including IQL (Kostrikov et al., 2021), CQL (Kumar et al., 2020), and BEAR (Kumar et al., 2019a); (iii) Imitation learning methods (IL) that do not explicitly encode reward or value information. Specifically, we consider Behavior Cloning (BC) (Pomerleau, 1988), its variant 10% BC which only uses 10% of trajectories with the highest return, and TD3+BC (Fujimoto et al., 2019) for comparison.

Since the labeled RTGs are informative enough about the "goodness" of actions, we implement Trifle by adopting the single-step value estimate (i.e., $V_t = \mathrm{RTG}_t$). Moreover, we built Trifle on top of the TT (Janner et al., 2021) algorithm, i.e., we directly take $p_{\mathrm{TT}}(a_t|s_t)$ derived by TT as our prior policy $p_{\mathrm{GPT}}(a_t|s_t)$ in Equation (2). See Appx. D.1 for more algorithmic details.

**Empirical Insights** Results are shown in Table 1. First, Trifle achieves the most state-of-the-art scores and especially performs well in Med-Replay datasets[6] which consist of much more suboptimal trajectories compared to the two other types of datasets. Next, compared to every individual baseline, Trifle achieves better results in 5 to 9 out of 9 benchmarks. To further examine the benefit brought by TPMs, we compare Trifle with TT since in this setup, their main algorithmic difference is the use of the improved proposal distribution (Eq. 2) for sampling actions. We can see that Trifle outperforms TT by a large margin in all environments, indicating that Trifle can robustly sample better actions from the high-dimensional action space.[7]

### 6.2 A STOCHASTIC ENVIRONMENT: THE MODIFIED GYM-TAXI ENVIRONMENT

This section further challenges Trifle on stochastic environments with highly suboptimal trajectories as well as labeled RTGs in the offline dataset. As demonstrated in Section 3, in this case, it is even hard to obtain accurate value estimates due to the stochasticity of transition dynamics. Section 4.2 demonstrates the potential of Trifle to more reliably estimate and sample action sequences under suboptimal labeled RTGs and stochastic environments. This section examines this claim by comparing the four following algorithms: (i) Trifle that adopts $V_t = \mathrm{RTG}_t$ (termed single-step Trifle or **s-Trifle**);

---

[6]Trifle achieves the highest scores averaged over 3 Med-Replay datasets.

[7]To perform more comprehensive evaluation, we also implement Trifle on top of the DT algorithm by replacing $p_{\mathrm{GPT}}(a_t|s_t)$ with $p_{\mathrm{DT}}(a_t|s_t)$, the results are discussed in Appx. E.

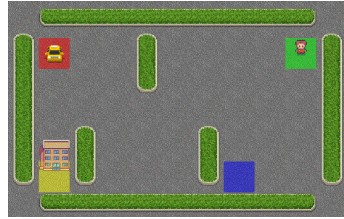

| Methods | Episode return | # penalty | $P(\text{failure})$ |
|---|---|---|---|
| s-Trifle | -99 | **0.14** | 0.11 |
| m-Trifle | **-57** | 0.38 | **0.02** |
| TT | -182 | 2.57 | 0.34 |
| DT | -388 | 14.2 | 0.66 |
| dataset | -128 | 2.41 | 0 |

(a) The stochastic Taxi environment.     (b) Empirical performance on the stochastic Taxi environment.

Figure 3: Results on the stochastic Taxi environment. All the reported numbers are averaged over 1000 trials. #penalty denotes the average number of penalties for executing illegal actions in an episode, and $P(\texttt{failure})$ denotes the probability of failing to transport the passenger to the destination.

(ii) Trifle equipped with $V_t = \sum_{\tau=t}^{t'} r_\tau + \text{RTG}_{t'}$ (termed multi-step Trifle or **m-Trifle**);[8] (iii) Trajectory Transformers (TT) (Janner et al., 2021); (iv) Decision Transformers (DT) (Chen et al., 2021). Among the four algorithms, s-Trifle and DT do not compute the "more accurate" multi-step value, and TT approximates the value by Monte Carlo samples. Therefore, we expect their relative performance to be DT $\approx$ s-Trifle $<$ TT $<$ m-Trifle.

**Environment setup** We create a stochastic variant of the Gym-Taxi Environment (Dieterich, 2000). As shown in Figure 3a, a taxi resides in a grid world consisting of a passenger and a destination. The taxi is tasked to first navigate to the passenger's position and pick them up, and then drop them off at the destination. There are 6 discrete actions available at every step: (i) 4 navigation actions (North, South, East, or West), (ii) Pick-up, (iii) Drop-off. Whenever the agent attempts to execute a navigation action, it has $0.3$ *probability of moving toward a randomly selected unintended direction*. At the beginning of every episode, the taxi, the passenger, and the destination are randomly initialized in one of 25, 5, and 4 locations, respectively. The reward function is defined as follows: (i) -1 for each action undertaken; (ii) an additional +20 for successful passenger delivery; (iii) -4 for hitting the walls; (iv) -5 for hitting the boundaries; (v) -10 for executing Pick-up or Drop-off actions unlawfully (e.g., executing Drop-off when the passenger is not in the taxi).

Following the Gym-MuJoCo benchmarks, we collect offline trajectories by running a Q-learning agent (Watkins & Dayan, 1992) in the above environment and recording the first 1000 trajectories that successfully drop off the passenger at the desired location.

**Empirical Insights** We first examine the accuracy of estimated returns for s-Trifle, m-Trifle, and TT. DT is excluded since it does not explicitly estimate the value of action sequences. Figure 2 illustrates the correlation between predicted and ground-truth returns of the three methods. First, s-Trifle performs the worst since it merely uses the inaccurate $\text{RTG}_t$ to approximate the ground-truth return. Next, thanks to its ability to exactly compute the multi-step value estimates, m-Trifle outperforms TT, which approximates the multi-step value with Monte Carlo samples.

We proceed to evaluate their performance in the stochastic Taxi environment. Besides the episode return, we adopt two metrics to better evaluate the adopted methods: (i) #penalty: the average number of executing illegal actions within an episode; (ii) $P(\texttt{failure})$: the probability of failing to transport the passenger within 300 steps. As shown in Figure 3b, the relative performance of the four algorithms is DT $<$ TT $<$ s-Trifle $<$ m-Trifle, which largely aligns with the anticipated results. The only "surprising" result is the superior performance of s-Trifle compared to TT. One plausible explanation for this behavior is that while TT can better estimate the given actions, the inferior performance is caused by its inability to efficiently sample rewarding actions.

### 6.3 ACTION-SPACE-CONSTRAINED GYM-MUJOCO VARIANTS

This section demonstrates that Trifle can be readily extended to safe RL tasks thanks to the TPM's ability to compute conditional probabilities. Specifically, besides achieving high expected returns, safe RL tasks require additional constraints on the action or future states to be satisfied. Therefore, define the constraint as $c$, our goal is to sample actions from $p(a_t | s_t, \mathbb{E}[V_t] \geq v, c)$, which can be achieved by additionally conditioning on $c$ in the candidate action sampling process.

---

[8]Please refer to Appx. D.2 for additional details of m-Trifle.

Table 2: Normalized Scores on the Action-Space-Constrained Gym-MuJoCo Variants. The results of Trifle and TT are both averaged over 12 random seeds, with mean and standard deviations reported.

| Dataset | Environment | Trifle | TT |
|---------|-------------|--------|-----|
| Med-Expert | Halfcheetah | **81.9**$_{\pm 4.8}$ | 77.8$_{\pm 5.4}$ |
| Med-Expert | Hopper | **109.6**$_{\pm 2.4}$ | 100.0$_{\pm 4.2}$ |
| Med-Expert | Walker2d | **105.1**$_{\pm 2.3}$ | 103.6$_{\pm 4.9}$ |

**Environment setup**  In MuJoCo environments, each dimension of $a_t$ represents the torque applied on a certain rotor of the hinge joints at timestep $t$. We consider action space constraints in the form of "value of the torque applied to the foot rotor $\leq A$", where $A = 0.5$ is a threshold value, for three MuJoCo environments: Halfcheetah, Hopper, and Walker2d. Note that there are multiple foot joints in Halfcheetah and Walker2d, so the constraint is applied to multiple action dimensions.[9] For all settings, we adopt the "Medium-Expert" offline dataset as introduced in Section 6.1.

**Empirical Insights**  The key challenge in these action-constrained tasks is the need to account for the constraints applied to other action dimensions when sampling the value of some action variable. For example, autoregressive models cannot take into account constraints added to variable $a_t^{i+1}$ when sampling $a_t^i$. Therefore, while enforcing the action constraint is simple, it remains hard to simultaneously guarantee good performance. As shown in Table 2, owing to its ability to exactly condition on the action constraints, Trifle outperforms TT significantly across all three environments.

## 7 RELATED WORK AND CONCLUSION

In offline RL tasks, our goal is to utilize a dataset collected by unknown policies to derive an improved policy without further interactions with the environment. Under this paradigm, we wish to generalize beyond naive imitation learning and stitch good parts of the behavior policy. To pursue such capabilities, many recent works frame offline RL tasks as conditional modeling problems that generate actions with high expected returns (Chen et al., 2021; Ajay et al., 2022; Ding et al., 2023) or its proxies such as immediate rewards (Kumar et al., 2019b; Schmidhuber, 2019; Srivastava et al., 2019). Recent advances in this line of work can be highly credited to the powerful expressivity of modern sequence models, since by accurately fitting past experiences, we can obtain 2 types of information that potentially imply high expected returns: (i) transition dynamics of the environment, which serves as a necessity for planning in model-based fashion (Chua et al., 2018), (ii) a decent policy prior which act more reasonably than a random policy to improve from (Janner et al., 2021).

While prior works on model-based RL (MBRL) also leverage models of the transition dynamics and the reward function (Kaiser et al., 2019; Heess et al., 2015; Amos et al., 2021), the above-mentioned RvS approaches focus more on directly modeling the correlation between actions and their end-performance. Specifically, MBRL approaches focus on planning *only* with the environment model. Despite being theoretically appealing, MBRL requires heavy machinery to account for the accumulated errors during rollout (Jafferjee et al., 2020; Talvitie, 2017) and out-of-distribution problems (Zhao et al., 2021; Rigter et al., 2022). All these problems add a significant burden on the inference side, which makes MBRL algorithms less appealing in practice. In contrast, while RvS algorithms can mitigate this inference-time burden by directly learning the correlation between actions and returns, the suboptimality of labeled returns could significantly degrade their performance. One potential solution is by combining RvS algorithms with temporal-difference learning methods that can correct errors in the labeled returns (Zheng et al., 2022; Yamagata et al., 2023).

While also aiming to mitigate the problem caused by suboptimal labeled RTGs, our work takes a substantially different route — by leveraging TPMs to mitigate the inference-time computational burden (e.g., by efficiently computing the multi-step estimates). Specifically, we identified two major problems that are caused by the lack of tractability in the sequence models: one regarding estimating expected returns and the other for conditionally sampling actions. We show that with the ability to compute more queries efficiently, we can partially solve both identified problems. In summary, this work provides positive evidence of the potential benefit of tractable models on RvS algorithms, and encourages the development of more inference-aware RvS methods.

---

[9]We only add constraints to the front joints in the Halfcheetah environment since the performance degrades significantly for all methods if the constraint is added to all foot joints.

## 8 REPRODUCIBILITY STATEMENT

Complete proofs of the theoretical results are provided in Appx. A. For the empirical results, the model architectures and hyperparameters are documented in Appx. C.3 and Appx. D. We plan to release our code as open source.

**Acknowledgements** This work was funded in part by the DARPA PTG Program under award HR00112220005, the DARPA ANSR program under award FA8750-23-2-0004, NSF grants #IIS-1943641, #IIS-1956441, #CCF-1837129, and a gift from RelationalAI. GVdB discloses a financial interest in RelationalAI.

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

# Supplementary Material

## A  PROOF OF THEOREM 1

To improve the clarity of the proof, we first simplify the notations in Thm. 1: define $\mathbf{X}$ as the boolean action variables $a_t := \{a_t^i\}_{i=1}^k$, and $Y$ as the variable $V_t$, which is a categorical variable with two categories 0 and 1. We can equivalently interpret $Y$ as a boolean variable where the category 0 corresponds to F and 1 corresponds to T. Dropping the condition on $s_t$ everywhere for notation simplicity, we have converted the problem into the following one:

Assume boolean variables $\mathbf{X} := \{X_i\}_{i=1}^k$ and $Y$ follow a Naive Bayes distribution: $p(\boldsymbol{x}, y) := p(y) \cdot \prod_i p(x_i|y)$. We want to prove that computing $p(\boldsymbol{x}|\mathbb{E}[y] \geq v)$, which is defined as follows, is NP-hard.

$$p(\boldsymbol{x}|\mathbb{E}[y] \geq v) := \frac{1}{Z} \begin{cases} p(\boldsymbol{x}) & \text{if } \mathbb{E}_{y \sim p(\cdot|\boldsymbol{x})}[y] \geq v, \\ 0 & \text{otherwise.} \end{cases} \tag{3}$$

By the definition of $Y$ as a categorical variable with two categories 0 and 1, we have

$$\mathbb{E}_{y \sim p(\cdot|\boldsymbol{x})}[y] = p(y = \mathtt{T}|\boldsymbol{x}) \cdot 1 + p(y = \mathtt{F}|\boldsymbol{x}) \cdot 0 = p(y = \mathtt{T}|\boldsymbol{x}).$$

Therefore, we can rewrite $p(\boldsymbol{x}|\mathbb{E}[y] \geq v)$ as

$$p(\boldsymbol{x}|\mathbb{E}[y] \geq v) := \frac{1}{Z} \cdot p(\boldsymbol{x}) \cdot \mathbb{1}[p(y = \mathtt{T}|\boldsymbol{x}) \geq v],$$

where $\mathbb{1}[\cdot]$ is the indicator function. In the following, we show that computing the normalizing constant $Z := \sum_{\boldsymbol{x}} p(\boldsymbol{x}) \cdot \mathbb{1}[p(y = \mathtt{T}|\boldsymbol{x}) \geq v]$ is NP-hard by reduction from the number partition problem, which is a known NP-hard problem. Specifically, for a set of $k$ numbers $n_1, \ldots, n_k$ ($\forall i, n_i \in \mathbb{Z}^+$), the number partition problem aims to decide whether there exists a subset $S \subseteq [k]$ (define $[k] := \{1, \ldots, k\}$) that partition the numbers into two sets with equal sums: $\sum_{i \in S} n_i = \sum_{j \notin S} n_j$.

For every number partition problem $\{n_i\}_{i=1}^k$, we define a corresponding Naive Bayes distribution $p(\boldsymbol{x}, y)$ with the following parameterization: $p(y = \mathtt{T}) = 0.5$ and[10]

$$\forall i \in [k], \; p(x_i = \mathtt{T}|y = \mathtt{T}) = \frac{1 - e^{-n_i}}{e^{n_i} - e^{-n_i}} \; \text{ and } \; p(x_i = \mathtt{T}|y = \mathtt{F}) = e^{n_i} \cdot \frac{1 - e^{-n_i}}{e^{n_i} - e^{-n_i}}.$$

It is easy to verify that the above definitions lead to a valid Naive Bayes distribution. Further, we have

$$\forall i \in [k], \; \log \frac{p(x_i = \mathtt{T}|y = \mathtt{T})}{p(x_i = \mathtt{T}|y = \mathtt{F})} = n_i \; \text{ and } \; \log \frac{p(x_i = \mathtt{F}|y = \mathtt{T})}{p(x_i = \mathtt{F}|y = \mathtt{F})} = -n_i. \tag{4}$$

We pair every partition $S$ in the number partition problem with an instance $\boldsymbol{x}$ such that $\forall i, x_i = \mathtt{T}$ if $i \in S$ and $x_i = \mathtt{F}$ otherwise. Choose $v = 2/3$, the normalizing constant $Z$ can be written as

$$Z = \sum_{\boldsymbol{x} \in \mathsf{val}(\mathbf{X})} p(\boldsymbol{x}) \cdot \mathbb{1}\big[p(y = \mathtt{T}|\boldsymbol{x}) \geq 2/3\big]. \tag{5}$$

Recall the one-to-one correspondence between $S$ and $\boldsymbol{x}$, we rewrite $p(y = \mathtt{T}|\boldsymbol{x})$ with the Bayes formula:

$$p(y = \mathtt{T}|\boldsymbol{x}) = \frac{p(y = \mathtt{T}) \prod_i p(x_i|y = \mathtt{T})}{p(y = \mathtt{T}) \prod_i p(x_i|y = \mathtt{T}) + p(y = \mathtt{F}) \prod_i p(x_i|y = \mathtt{F})},$$

$$= \frac{1}{1 + e^{-\sum_i \log \frac{p(x_i|y=\mathtt{T})}{p(x_i|y=\mathtt{F})}}},$$

$$= \frac{1}{1 + e^{-(\sum_{i \in S} n_i - \sum_{j \notin S} n_j)}},$$

where the last equation follows from Equation (4). After some simplifications, we have

$$\mathbb{1}\big[p(y = \mathtt{T}|\boldsymbol{x}) \geq 2/3\big] = \mathbb{1}\big[\sum_{i \in S} n_i - \sum_{j \notin S} n_j \geq 1\big].$$

---

[10]Note that we assume the naive Bayes model is parameterized using log probabilities.

Plug back to Equation (5), we have

$$Z = \sum_{S \subseteq [k]} p(\boldsymbol{x}) \cdot \mathbb{1}\big[\sum_{i \in S} n_i - \sum_{j \notin S} n_j \geq 1\big],$$

$$= \frac{1}{2} \sum_{S \subseteq [k]} p(\boldsymbol{x}) \cdot \mathbb{1}\big[\sum_{i \in S} n_i - \sum_{j \notin S} n_j \neq 0\big],$$

where the last equation follows from the fact that (i) if $\boldsymbol{x}$ satisfy $\sum_{i \in S} n_i - \sum_{j \notin S} n_j \geq 1$ then $\bar{\boldsymbol{x}}$ has $\sum_{i \in S} n_i - \sum_{j \notin S} n_j \leq -1$ and vise versa, and (ii) $\sum_{i \in S} n_i - \sum_{j \notin S} n_j$ must be an integer.

Note that for every solution $S$ to the number partition problem, $\sum_{i \in S} n_i - \sum_{j \notin S} n_j = 0$ holds. Therefore, there exists a solution to the defined number partition problem if $Z < \frac{1}{2}$. □

## B TRACTABLE PROBABILISTIC MODELS (TPM)

A probabilistic model can be seen as a black box to answer queries about the quantities of interest of the joint probability distribution, such as computing marginal probability and performing maximum-a-posterior inference given some evidence. Tractable probabilistic models provide more guarantees when answering probabilistic queries: (i) it can perform exact inference to the model's distribution and no approximations are required. (ii) the query computation can be carried out efficiently, that is, in time polynomial (linear in many cases) in the size of the model. Notably, tractability is defined for a family of models only w.r.t. a class of queries and not an absolute property. Indeed, a tractable representation for one query class might not admit polynomial time inference for another query class.

## C PROBABILISTIC CIRCUITS (PC)

### C.1 DEFINITION OF PC

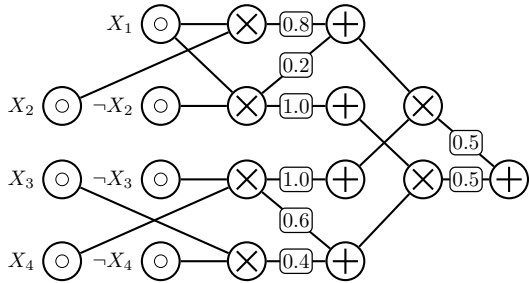

Figure 4: An example PC.

Probabilistic circuits (PCs) represent a wide class of TPMs that model probability distributions with a parameterized directed acyclic computation graph (DAG). Specifically, a PC $p(\mathbf{X})$ defines a joint distribution over a set of random variables $\mathbf{X}$ by a single root node $n_r$. A PC contains three kinds of computational nodes: *input*, *sum*, and *product*. As illustrated in Figure 4, the example PC in Figure 4 defines a joint distribution over 4 random variables $X_1, X_2, X_3, X_4$. Each leaf node in the DAG serves as an *input* node that encodes a univariate distribution (e.g., Guassian, Categorical), while *sum* nodes or *product* nodes are inner nodes, distinguished by whether they are doing mixture or factorization over their child distributions (denoted $\mathsf{in}(n)$). A PC defines a probability distribution in the following recursive way:

$$p_n(\boldsymbol{x}) := \begin{cases} f_n(\boldsymbol{x}) & \text{if } n \text{ is an input unit,} \\ \sum_{c \in \mathsf{in}(n)} \theta_{n,c} \cdot p_c(\boldsymbol{x}) & \text{if } n \text{ is a sum unit,} \\ \prod_{c \in \mathsf{in}(n)} p_c(\boldsymbol{x}) & \text{if } n \text{ is a product unit,} \end{cases}$$

where $\theta_{n,c}$ represents the parameter corresponding to edge $(n, c)$ in the DAG. For sum units, we have $\sum_{c \in \mathsf{in}(n)} \theta_{n,c} = 1, \theta_{n,c} \geq 0$, and we assume w.l.o.g. that a PC alternates between the sum and product layers before reaching its inputs.

## C.2 TRACTABILITY AND EXPRESSIVITY

**Definition 1** (Decomposability). A PC is decomposable if for every product unit $n$, its children have disjoint scopes:

$$\forall c_1, c_2 \in \mathsf{in}(n) \; (c_1 \neq c_2), \; \phi(c_1) \cap \phi(c_2) = \varnothing.$$

**Definition 2** (Smoothness). A PC is smooth if for every sum unit $n$, its children have the same scope:

$$\forall c_1, c_2 \in \mathsf{in}(n), \; \phi(c_1) = \phi(c_2).$$

Similar to neural networks, a large PC contains millions of computation units which enables expressive modeling. However, to guarantee the desired tractability, i.e., the ability to answer numerous probabilistic queries, some properties have to be imposed on their DAG structures. For instance, *smoothness* together with *decomposability* ensure that a PC can compute arbitrary marginal/conditional probabilities in linear time w.r.t. its size, i.e., the number of edges in its DAG. These are properties of the variable scope $\phi(n)$ of PC unit $n$, that is, the variable set comprising all its descendent nodes. Here we introduce how we execute marginalization inference in practice on a decomposable and smooth PC. Take the example PC in Figure 4 as an example, if we want to acquire $p_n(X_1, X_2, X_3)$ by marginalizing out $X_4$, we only need to set the output probability of $X_4$'s input nodes to 1 and follow the same forward process as computing $p_n(X_1, X_2, X_3, X_4)$.

Notably, such structural constraints raise higher requirements for PC optimizers to acquire a comparably expressive PC. Thankfully, recent advancement achieved in PC modeling largely bridges this gap and makes it possible to model distributions of more complex data including the offline RL data.

## C.3 TRAINING DETAILS OF THE ADOPTED PC

### C.3.1 THE EM PARAMETER LEARNING ALGORITHM

As mentioned above, a PC takes as input a sample $x$ and outputs the corresponding probability $p_n(x)$. Given a dataset $\mathcal{D}$, the PC optimizer takes the PC parameters (consisting of sum edge parameters and input node/distribution parameters) as input and aims to maximize the MLE objective $\sum_{x \in \mathcal{D}} \log p_n(x)$. Since PCs can be deemed as latent variable models with hierarchically nested latent space, the Expectation-Maximization (EM) algorithm is usually the default choice for PC parameter learning. Specifically, considering gradient-based EM, the feedforward computation of PCs which involves computing the log-likelihood $\log p_n(x)$ is differentiable and can be modeled by a computation graph. Therefore, we can efficiently compute its gradient w.r.t. each parameter via the backpropagation algorithm. Given a mini-batch of samples, the backpropagation algorithm is used to accumulate gradients for each parameter.

### C.3.2 TRAINING PC WITH QUANTILE-DISCRETIZED MUJOCO DATASET

For the PC implemented in Trifle, we adopt the Hidden Chow-Liu Tree (HCLT) structure (Liu & Van den Broeck, 2021) and categorical distributions for its input nodes. We use the same quantile dataset discretized from the original Gym-MuJoCo dataset as TT, where each raw continuous variable is divided into 100 categoricals, and each categorical represents an equal amount of probability mass under the empirical data distribution (Janner et al., 2021).

During training, we first derive an HCLT structure given the data distribution and then utilize the latent variable distillation technique (LVD) to do parameter learning (Liu et al., 2022). Specifically, the neural embeddings used for LVD are acquired by a BERT-like Transformer (Devlin et al., 2018) trained with the Masked Language Model task. To acquire the embeddings of a subset of variables $\phi$, we feed the Transformer with all other variables and concatenate the last Transformer layer's output for the variables $\phi$. Please refer to the original paper for more details.

## D ADDITIONAL ALGORITHMIC DETAILS OF TRIFLE

### D.1 GYM-MUJOCO

**Sampling Details**. We take the single-step value estimate by setting $V_t = \mathrm{RTG}_t$ and sample $a_t$ from Equation (2). When training the GPT used for querying $p_{GPT}(a_t^i | s_t, a_t^{<i})$, we adopt the same

model specification and training pipeline as TT. When computing $p_{TPM}(V_t \geq v|s_t, a_t^{\leq i})$, we first use the learned PC to estimate $p(V_t|s_t)$ by marginalizing out intermediate actions $a_{t:t'}$ and select the $\epsilon$-quantile value of $p(V_t|s_t)$ as our prediction threshold $v$ for each inference step. Empirically we fixed $\epsilon$ for each environment and $\epsilon$ ranges from 0.1 to 0.3.

**Beam Search Hyperparameters**. The maximum beam width $N$ and planning horizon $H$ that Trifle uses across 9 MuJoCo tasks are 15 and 64, respectively.

### D.2 STOCHASTIC TAXI ENVIRONMENT

Except for s-Trifle, the sequence length $K$ modeled by TT, DT, and m-Trifle is all equal to 7. The inference algorithm of TT follows that of the MuJoCo experiment and DT follows its implementation in the Atati benchmark. Notably, during evaluation, we condition the pretrained DT on 6 different RTGs ranging from -100 to -350 and choose the best policy resulting from RTG=-300 to report in Figure 3b. Beam width $N = 8$ and planning horizon $H = 3$ hold for TT and m-Trifle.

## E TRIFLE BUILT ON DECISION TRANSFORMER

Table 3: Normalized Scores on the Standard Gym-MuJoCo Benchmarks.

| Dataset | Environment | Trifle | DT |
|---------|-------------|--------|-----|
| Med-Expert | Halfcheetah | **91.9**$_{\pm1.9}$ | 86.8$_{\pm1.3}$ |
| Medium | Halfcheetah | **44.2**$_{\pm0.7}$ | 42.6$_{\pm0.1}$ |
| Med-Replay | Halfcheetah | **39.2**$_{\pm0.4}$ | 36.6$_{\pm0.8}$ |
| Med-Expert | Walker2d | **108.6**$_{\pm0.3}$ | 108.1$_{\pm0.2}$ |
| Medium | Walker2d | **81.3**$_{\pm2.3}$ | 74$_{\pm1.4}$ |

We choose TT as our base policy for its simple yet effective nature as well as its competitive performance. However, we highlight that by replacing $p_{GPT}(a_t|s_t)$ (cf. Equation (1)) with better prior distributions (generated by other RvS algorithms), Trifle could still yield a decent performance gain. To verify this, we adopt Trifle on top of the DT algorithm by replacing $p_{GPT}(a_t|s_t)$ with $p_{DT}(a_t|s_t)$ and run experiments on the Gym-Mujoco benchmarks. Results are provided in Table 3

As shown in Table 3, adding the TPM correction term to DT could also boost DT's performance, which lives up to our expectations. Specifically, the architecture and hyperparameters of Trifle based on DT follow that of Trifle built on TT, while the results of DT are obtained from the original paper (we adopt the same number of seeds as the DT paper).

## F STANDARD DEVIATION OF ALL RVS BASELINES

Table 4: Standard deviation of all RvS baselines.

| Dataset | Environment | Trifle | BR-RCRL | DD | TT | DT |
|---------|-------------|--------|---------|-----|-----|-----|
| Med-Expert | Halfcheetah | 95.1$_{\pm0.3}$ | 95.2$_{\pm0.8}$ | 90.6$_{\pm1.3}$ | 95.0$_{\pm0.2}$ | 86.8$_{\pm1.3}$ |
| Med-Expert | Hopper | 113.0$_{\pm0.4}$ | 112.9$_{\pm0.9}$ | 111.8$_{\pm1.8}$ | 110.0$_{\pm2.7}$ | 107.6$_{\pm1.8}$ |
| Med-Expert | Walker2d | 109.3$_{\pm0.1}$ | 111.0$_{\pm0.4}$ | 108.8$_{\pm1.7}$ | 101.9$_{\pm6.8}$ | 108.1$_{\pm0.2}$ |
| Medium | Halfcheetah | 49.5$_{\pm0.2}$ | 48.6$_{\pm1.1}$ | 49.1$_{\pm1.0}$ | 46.9$_{\pm0.4}$ | 42.6$_{\pm0.1}$ |
| Medium | Hopper | 67.1$_{\pm4.3}$ | 78.0$_{\pm1.3}$ | 79.3$_{\pm3.6}$ | 61.1$_{\pm3.6}$ | 67.6$_{\pm1.0}$ |
| Medium | Walker2d | 83.1$_{\pm0.8}$ | 82.3$_{\pm1.7}$ | 82.5$_{\pm1.4}$ | 79.0$_{\pm2.8}$ | 74.0$_{\pm1.4}$ |
| Med-Replay | Halfcheetah | 45.0$_{\pm0.3}$ | 42.3$_{\pm3.3}$ | 39.3$_{\pm4.1}$ | 41.9$_{\pm2.5}$ | 36.6$_{\pm0.8}$ |
| Med-Replay | Hopper | 97.8$_{\pm0.3}$ | 98.3$_{\pm2.6}$ | 100$_{\pm0.7}$ | 91.5$_{\pm3.6}$ | 82.7$_{\pm7.0}$ |
| Med-Replay | Walker2d | 88.3$_{\pm3.8}$ | 80.6$_{\pm2.5}$ | 75$_{\pm4.3}$ | 82.6$_{\pm6.9}$ | 66.6$_{\pm3.0}$ |

The standard deviation numbers for all RvS baselines are shown in Table 4. Trifle achieves relatively lower standard deviations than other baselines, indicating that Trifle performs more robustly. Notably,

the results of different baselines are averaged over different numbers of random seeds. In general, we evaluate Trifle on 12 random seeds. However, towards a more reliable evaluation, we improved the number of random seeds to 20 for tasks with relatively high standard deviation. The number of evaluation seeds for BR-RCRL, DD, TT, and DT are 4, 5, 15, and 3, respectively.

