# OpenReview forum: "Expressive Modeling is Insufficient for Offline RL: A Tractable Inference Perspective"
_ICLR.cc/2024/Conference — Submitted to ICLR 2024_

### Official Review · Reviewer_dbnt · 2023-10-30

**Soundness:** 3 good
**Presentation:** 4 excellent
**Contribution:** 4 excellent
**Rating:** 8
**Confidence:** 4

**Summary:**

The authors examine the problem of offline RL via RvS approaches, and note that one (underexamined) issue is tractability - answering flexible probabilistic queries faithfully. Due to the nature of the dataset generation and stochastic MDP structure, this is particularly difficult. The authors use TPMs for RvS to address this problem, showing that they are better at achieving the requested returns, and perform competitively, particularly with suboptimal data. Furthermore, they are better suited for constrained-action RL.

**Strengths:**

This is a very well-written and clear paper. The authors do a very good job of concisely going through the material (offline RL and tractability), and motivate their solution with theory and empirical evidence. There is an extensive experimental section with different environments, many baselines, and various examinations of the results.

**Weaknesses:**

The improvement in scores over baselines don't seem so large. In particular, with constrained actions one might expect the TPM approach to produce more gains. However, I don't see this as a particular demerit for insightful research.

**Questions:**

I do not have any questions.

**Edit:** I have read the other reviews and the authors' feedback, and see no reason to update my (very positive) original rating.

---

> ### Author Response · Authors · 2023-11-20
> **Response to reviewer dbnt**
>
> Thank you for your constructive and encouraging feedback on our paper. We greatly appreciate your recognition of our work. Regarding the improvement in scores of Trifle over the baselines, we have explained the results in a more detailed way and added more empirical evidence, please refer to the "Response to All" part, thanks!

---

### Official Review · Reviewer_WTWy · 2023-10-30

**Soundness:** 3 good
**Presentation:** 3 good
**Contribution:** 3 good
**Rating:** 5
**Confidence:** 3

**Summary:**

The paper looks at the "tractability" (ability to answer probabilistic queries) issue in offline RL and uses TPMs (Tractable Probabilistic Models) for solving complex RL tasks.

**Strengths:**

1. The paper is very well written. I especially enjoyed reading section 4.

2. The experiments are well-designed and the baselines are good as well (though do note the comments in the next two sections).

**Weaknesses:**

The points are in the spirit of making the paper stronger and in some cases, did not contribute to the score.

1. It will be useful to have some details about the Probabilistic Circuits in. the main paper while the bulk of the details can be in appendix (as is the case now).

2. The experiments used medium (or medium-expert) datasets. It will be useful to see the behavior with expert only and weak datasets.

3. It is not clear if the proposed approach outperforms "Bayesian Reprameterized RCRL" baseline. While I do not think it is important to beat all the baselines, it is useful to (i) understand how the two methods stack against each other (sharing standard deviation numbers for BR-RCRL will help) and (ii) the difference between the two approaches. The authors should add more details about why they think the two approaches are tied so closely.

4. The authors should consider adding results with another TPM (even if for a subset of tasks) so that it is clear that their approach works across different TPMs.

5. It is not clear why the authors used the TT baseline (in place of a stronger alternative) for experiments in 6.2 and 6.3

**Questions:**

Listing some questions (to make sure I better understand the paper) and potential areas of improvement. Looking forward to engaging with the authors on these questions and the points in the weakness section

1. Are terms like "training time optimality" introduced by this paper? If yes, could we consider using existing terms like "expressivity"?
2. The authors mentioned a sampling approach where "we first sample from a prior distribution p(a_t | s_t ), and then reject actions with low expected returns" (section 4.1). Did they use this in conjunction with any of the baselines (to make them stronger) ?
3. Was beam-search used in the baselines ?
4. In table 1, could the authors report the standard deviation for all the approaches.

---

> ### Author Response · Authors · 2023-11-20
> **Response to the reviewer WTWy**
>
> ### **The reason why we compare Trifle with TT.**
> As mentioned above, in Sections 6.1 and 6.3, we select $p_\mathrm{TT}(a_t|s_t)$ as our prior policy $p_\mathrm{GPT}(a_t|s_t)$, so the performance gain compared with TT fairly demonstrates the significance of Trifle in improving inference-time optimality; In Section 6.2, we didn’t evaluate BR-RCRL for there is no official released code, and we consider comparison with TT and DT is sufficient for demonstrating the effectiveness of adopting TPM to handle environment stochasticity.
>
> ### **More details about Trifle compared to BR-RCRL.**
> | Dataset    | Environment | Trifle    | BR-RCRL   |
> | ---------- | ----------- | --------- | --------- |
> | Med-Expert | Halfcheetah | 95.1±0.3  | 95.2±0.8  |
> | Medium     | Halfcheetah | 113.0±0.4 | 112.9±0.9 |
> | Med-Replay | Halfcheetah | 109.3±0.1 | 111.0±0.4 |
> | Med-Expert | Hopper      | 49.5±0.2  | 48.6±1.1  |
> | Medium     | Hopper      | 67.1±4.3  | 78.0±1.3  |
> | Med-Replay | Hopper      | 83.1±0.8  | 82.3±1.7  |
> | Med-Expert | Walker2d    | 45.0±0.3  | 42.3±3.3  |
> | Medium     | Walker2d    | 97.8±0.3  | 98.3±2.6  |
> | Med-Replay | Walker2d    | 88.3±3.8  | 80.6±2.5  |
>
>
> 1. The standard deviation numbers for all RvS baselines are shown in Table 4 of the updated paper. Here we show the standard deviatios of BR-RCRL and Trifle for camparison. In 7 out of 9 mujoco tasks, Trifle achieves a much lower standard deviation than BR-RCRL, indicating that Trifle performs more robustly. Notably, the results of BR-RCRL are averaged over only 5 random seeds while we evaluate Trifle on 12 random seeds. Moreover, towards a more reliable evaluation, we improved the number of random seeds to 20 for tasks with relatively high standard deviation.
>
> 2. The goal of BR-RCRL is also to sample actions with a higher probability of achieving high RTGs. This work differs from ours in two key aspects: (a) BR-RCRL adopts energy-based models (EBMs) to approximately sample from $p(a | s, R)$ using iterative inference; due to the intractability of EBMs, this sampling process could suffer from slow mixing time under high-dimensional action space. In contrast, Trifle leverages TPMs to exactly compute $p(a_i | s, a_{<i}, R>v)$ ($v$ is a threshold), and then uses beam search/rejection sampling to approximate samples from $p(a_i | s, a_{<i}, E[R]>v)$. Note that since we approximate the expected-return-conditioned policy iteratively over every action dimension, the rejection rate would be moderate. (b) Once the training finishes, BR-RCRL is limited to approximately sample from $p(a | s, R)$. In contrast, Trifle can answer arbitrary marginal/conditional queries over $(s,a,R)$, naturally enabling its application in more challenging scenarios like stochastic environments or constrained policy generation. For example,  in stochastic environments, we utilize Trifle to compute multi-step value $p(V_t\geq v | s_t, a_{t:t'})$ by marginalizing intermediate states; We could also apply Trifle to generate constrained policy  $p(a_t|s_t, E(V_t)\geq v, c)$, where c represents arbitrary state/action constraints.
>
> ### **Adding results with other TPMs.**
> We chose PC as the adopted TPM because it can be more easily scaled up to complex datasets thanks to recent advances. Also, PCs’ ability to efficiently compute arbitrary marginal/conditional probabilities meets the need of Trifle. Any TPM with sufficient expressiveness and support answering queries required by Trifle can be used.
>
> ### **"Training time optimality" vs "Expressivity"**
> Yes, "training time optimality" can be interpreted as "expressivity", i.e. how well/accurately the model can fit the offline trajectories to recover the ground truth distribution. We have modified the paper to clarify this, please refer to the second paragraph of Section 3.
>
> ### **Questions about the rejection sampling / beam search.**
> We implement the rejection sampling process via beam search, which is used in all applicable baselines. Specifically, Trifle leverages TPMs to exactly compute $p(a_i | s, a_{<i}, R>v)$ ($v$ is a threshold), and then uses beam search/rejection sampling to approximate samples from $p(a_i | s, a_{<i}, E[R]>v)$. Note that since we approximate the expected-return-conditioned policy iteratively over every action dimension, the rejection rate would be moderate.
>
> ### **The standard deviation for all RvS approaches.**
> The standard deviation numbers for all RvS baselines are shown in Table 4 of the updated paper. Trifle achieves relatively lower standard deviations than other baselines, indicating that Trifle performs more robustly. Notably, the results of different baselines are averaged over different numbers of random seeds. In general, we evaluate Trifle on 12 random seeds. However, towards a more reliable evaluation, we improved the number of random seeds to 20 for tasks with relatively high standard deviation. The number of evaluation seeds for BR-RCRL, DD, TT, and DT are 4, 5, 15, and 3, respectively.

---

### Official Review · Reviewer_94tr · 2023-11-01

**Soundness:** 3 good
**Presentation:** 4 excellent
**Contribution:** 2 fair
**Rating:** 6
**Confidence:** 4

**Summary:**

The paper proposes to use tractable probabilistic models (TPM) in reinforcement learning via supervised learning (RvS) approaches such that the computation of the multi-step value estimate can be done in a more tractable manner (in replacement of the potentially high-variance Monte-Carlo estimate needed in normal autoregressive generative models). The authors showed that through thorough empirical analysis that obtaining the multi-step value estimate accurately and act according to it is crucial in achieving good performance at the inference time, and previous approaches have failed to do so to some extent. In contrast, the use of TPM readily addresses such issue and consequently results in performance improvement on a range of offline reinforcement learning tasks tested (nine original D4RL locomotion tasks, a modified gym-taxi task, and three action-constrained safe RL locomotion tasks).

**Strengths:**

The paper is very well-written with clear presentation of the method and informative empirical results with comparison to relevant baseline methods. The analysis (Section 3) of the correlation between inference-time optimality score (how well an action is selected based on the model's estimate of the return) and the actual return achieved is convincing, and it motivates the proposed TPM-based solution well.

To the best of my knowledge, this paper is the first that uses TPM in offline RL and the thorough empirical study (especially the analysis on the estimated returns vs. actual returns in Figure 1 and 2) brings insights on how useful TPM is in the context of RL/offline RL/RvS.

**Weaknesses:**

The main weakness of the paper is the lack of convincing evidence that the proposed algorithm Trifle can also bring significant performance benefits to offline RL tasks.
- The harder D4RL tasks are not evaluated (Section 6.1). The nine tasks evaluated are relatively saturated at the moment and it is hard to see much performance gain (as seen in Table 1) on top of existing approaches. It would be great if the authors could test the method on harder tasks such as antmaze tasks.
- The performance improvements on two of the three domains considered (two MuJoCo domains, in Sec 6.1 and Sec 6.3) are marginal. There does seem to be a descent performance gain on one of the custom task (on the gym-taxi environment presented in Sec 6.2) but it is not a standard task that people have evaluated on (which is fine, but a more comprehensive set of experiments would make a stronger case).

Other minor comments:
- I found Theorem 1 to be a bit out of place because showing a problem is NP-hard brings little information on how easy it is to approximate the solution, which is what people mostly care about in practice.
- I found the details of the action-space-constrained task (Sec 6.3) to be quite terse. How is the constraint being conditioned (is it a boolean variable or the threshold value discussed in the caption of Table 2?) How is it being incorporated into the TPM?

**Questions:**

N/A

---

> ### Author Response · Authors · 2023-11-20
> **Response to reviewer 94tr**
>
> ### **Theorem 1 seems to be a bit out of place.**
> We totally agree with the reviewer that showing NP-hardness brings little information on how easy it is to approximate the solution. In fact, the main goal of Theorem 1 is to inform us that we need to resort to approximations of $p(a_t | s_t, E[V_t]>v)$.
>
> ### **Details of the action-space-constrained task.**
> We consider action space constraints in the form of “value of the torque applied to the foot rotor ≤ A”, where A is a threshold value. We enforced the constraint on TT by masking the unsatisfied categories (each action category can be mapped to a continuous torque value according to the adopted discretization strategy, if the reconstructed value is larger than A,  we consider this category unsatisfied) of the categorical distribution $p_\mathrm{TT} (a_t | s_t)$, and sampling from the masked distribution must satisfy the constraint. As for TPM, to compute the conditional probability, PC will involve both forward and backward processes. Therefore, we tell the satisfied categories to PC’s input nodes of the constrained action variables and PC will incorporate such constraint message into its forward and backward paths.

---

> > ### Comment · Reviewer_94tr · 2023-11-22
> > **Thank you for the responses!**
> >
> > Thanks for providing the details on the action-space-constrained task and additional experiments on more D4RL locomotion tasks. I would encourage the authors to include the details of the action-space-constrained task in the paper.
> >
> > I have read all other reviews and the responses to them and I would like to increase my score to 6.

---

### Official Review · Reviewer_WNH3 · 2023-11-02

**Soundness:** 3 good
**Presentation:** 3 good
**Contribution:** 2 fair
**Rating:** 5
**Confidence:** 3

**Summary:**

The paper considers the problem of RL via sequence modeling, an offline RL paradigm given offline trajectories of the form (s, a, r, R) where R denotes the return-to-go. This is achieved by optimizing and sampling from p(a_{t: T} | s_t, E[V_t] >= v).

The authors highlight that there are two main challenges in training such policies. (1) Estimation accuracy: Ability to estimate the expected return of a state and a corresponding action sequence. (2) Tractability issues: Ability to efficiently sample from this distribution. The authors claim that both of these issues are jointly responsible for poor test time performance of RvS based methods.

While fixing the estimation accuracy part is already an active area of research, the authors focus on fixing the tractability part by using Tractable Probabilistic Models (TPMs) for sampling P(V_t >= v | s_t, a_t). The authors experimentally show that their method works well in practice and beats many other offline RL methods on various benchmarks.

**Strengths:**

1. Address the tractability issue in RvS which seems to have been overlooked in the past works.
2. Incorporate TPMs into RL via sequence modeling.
3. Extensive experimental evaluation.

**Weaknesses:**

1. Limited intuition on what the TPM is doing. I would have appreciated if the authors can incorporate a section on what TPMs are and what they are designed to do. While I do see some discussion in Appendix B.1. on Probabilistic Circuits, a colloquially accessible introduction to TPMs in the main body is highly appreciated.

2. Triffle does not seem to be significantly better than other RvS algorithms.

**Questions:**

1. While the authors show in Theorem 1 that when |A| = 2^K, solving the sampling issue given the Naive Bayes Distribution could be NP-hard, can you please discuss why for this setting, sampling using (2) is efficient even when given oracle access to P_{GPT} and P_{TPM}? Can the authors discuss why computing Z or sampling using 2 is efficient when |A| = 2^K?

2. It is not clear from Table 1 and Table 2 if Trifle is significantly better than other RvS approaches. Can you please discuss any concrete examples where Trifle significantly (with a reasonable margin) outperforms other RvS approaches?

---

> ### Author Response · Authors · 2023-11-20
> **Response to reviewer WNH3**
>
> ### **Questions concerning the Theorem 1.**
> While Theorem 1 suggests computing $p(a_t | s_t,E[V_t]>v)$  is NP-hard, equation (2) is computing $p(a_t | s_t, V_t>v)$ as an approximation to $p(a_t | s_t, E[V_t]>v)$ (Note that there is an expectation operator). But, intuitively, incorporating the value information into the sampling phase could give us better samples than naive sampling from $p(a_t|s_t)$, where "better" means close to samples drawn from the desired distribution $p(a_t | s_t, E[V_t]>v)$. Moreover, computing (2) for each dimension $a_t^i$ is also non-trivial when $|A| = 2^K$ as we have to marginalize out unseen action variables $a_t^{i+1},...,a_t^k$ where $|A|=n^K$. This is why non-tractable models may suffer exponential complexity and we need the TPM's ability to compute arbitrary marginals in linear time.

---

### Author Response · Authors · 2023-11-20
**Response to all reviewers**

### **Introduction to the TPM and PC.**

We thank the reviewers for their valuable feedback and agree that including a section on TPMs and PCs in the main body of the paper would be beneficial. In the following, we provide a detailed elaboration on the fundamental concepts behind TPMs and PCs as well as the training algorithm of PC. **Due to space constraints, we added a concise paragraph to Section 2：**

- *Probabilistic circuits (PCs) are a general and unified computational framework for tractable probabilistic modeling, i.e. a wide class of TPMs can be represented as a PC. Similar to neural networks, PCs are also computation graphs containing millions of computation units, where constraints are imposed on their structures to enable efficient computation of various probabilistic queries. Generally, computing probabilistic queries involves executing specific feedforward/backward algorithms over the PC-defined computation graph.*


**and put the full explanation in Appendix B and Appendix C.** Concretely, we added the following contents:

- *B Tractable Probabilistic Models*

    *A probabilistic model can be seen as a black box to answer queries about the quantities of interest of the joint probability distribution, such as computing marginal probability and performing maximum-a-posterior inference given some evidence. Tractable probabilistic models provide more guarantees when answering probabilistic queries: i) it can perform exact inference to the model’s distribution and no approximations are required. ii) the query computation can be carried out efficiently, that is, in time polynomial (linear in many cases) in the size of the model. Notably, tractability is defined for a family of models only w.r.t. a class of queries and not an absolute property. Indeed, a tractable representation for one query class might not admit polynomial time inference for another query class.*

- *C.1 We provide an example PC in Figure 4 to illustrate how PC encodes a joint distribution.*

- *C.2 We use the example PC to show how we execute marginalization inference in practice.*

- *C.3.1 We introduce the Expectation-Maximization parameter learning algorithm of PCs.*

### **The performance of Trifle.**

To better understand the significance of Trifle, we would like to draw the reviewer's attention to the improvements of Trifle compared to TT. That’s because empirically we directly adopt $p_\mathrm{TT} (a_t | s_t)$ as our prior policy $p_\mathrm{GPT} (a_t | s_t)$ in Equation (1), and the large performance gain over TT indicates that the correction term $p_\mathrm{TPM}(V_t>v|s_t, a_t)$ can enhance the inference-time optimality of base policy reliably.

We choose TT as our base policy for its simple yet effective nature as well as its competitive performance. However, we highlight that by replacing $p_\mathrm{GPT} (a_t | s_t)$ (cf. Equation (1)) with better prior distributions (generated by other RvS algorithms), Trifle could still yield a decent performance gain.  To verify this, we adopt Trifle on top of the DT algorithm by replacing $p_\mathrm{GPT} (a_t | s_t)$ with $p_\mathrm{DT}(a_t|s_t)$ and run experiments on the Gym-Mujoco benchmarks. Results are provided in Table 3 of the updated paper and are copied below for your convenience. (We did not choose BR-RCRL because no official released code was found.)


| Dataset    | Environment | Trifle        | DT        |
| ---------- | ----------- | ------------- | --------- |
| Med-Expert | Halfcheetah | **91.9±1.9**  | 86.8±1.3  |
| Medium     | Halfcheetah | **44.2±0.7**  | 42.6±0.1  |
| Med-Replay | Halfcheetah | **39.2±0.4**  | 36.6±0.8  |
| Med-Expert | Walker2d    | **108.6±0.3** | 108.1±0.2 |
| Medium     | Walker2d    | **81.3±2.3**  | 74.0±1.4  |

As shown in the table, adding the TPM correction term to DT could also boost DT’s performance, which lives up to our expectations. Specifically, the architecture and hyperparameters of Trifle based on DT follow that of Trifle built on TT, while the results of DT are obtained from the original paper (we adopt the same number of seeds as the DT paper). Notably, we only report 5 results for the reason we have to modify the output layer of DT to make it combinable with TPM. Specifically, the original DT directly predicts deterministic action while the modified DT outputs categorical action distributions like TT. In the other unreported 4 environments, the modified DT fails to achieve the original DT scores, and we haven’t figured out the causes since the time is tight.

---

### Meta-Review · Area_Chair_pYyC · 2023-12-10

**Metareview:**

This paper studies reward-conditioned reinforcement learning, and highlights two problems with methods in this area: (1) estimation accuracy: ability to estimate the expected return of a state and a corresponding action sequence. (2) tractability issues: ability to efficiently sample from this distribution. The paper aims to tackle specifically the second problem by using tractable probabilistic models.

The reviewers liked the usage of TPMs, though there are concerns about evaluating on more complex tasks, such as Antmaze tasks, which were not studied in this paper (though they are a good case of Scenario #2 discussed in the paper). I also think the comparisons lack several other comparisons for an understanding of the approach: rather than running Equation 2 at inference time, why not train the distribution of action tokens with a loss that biases it towards high value? This is a method previously studied in the reward-conditioned policies paper, and regardless, would help us understand the tradeoff between estimation and inference errors.

The paper lacks baselines for instance the DoC baseline (https://openreview.net/references/pdf?id=C0SfmDkou) and other similar methods, that learn latent variables to address issues with DT and TT with stochastic environments. A comparison would be important to have in my opinion.

I also think that using Q-functions or advantage functions (for instance, the "TT+Q" method from the TT paper) should be compared to. To what extent such a better, well estimated RTG values (obtained in the form of Q or advantage functions) remove the need for inference-time tractability? These kinds of questions are important to understand the benefits of this approach in my opinion, especially given that on the tasks studied, the proposed method is not doing singnificantly better than prior approaches.

Overall I think while the ideas of changing inference is interesting and novel, it remains unclear if the method is really improving performance. Without a complete comparisons to other approaches (including those that bias the sampling of RvS type policies using estimates of Q-values or advantages at train time and those which handle stochasticity by learning other conditional variables for TT and DT), it is unclear if this paper can be accepted at this time.

**Justification For Why Not Higher Score:**

The. paper misses several important comparisons and the results as it is not solid enough to justify acceptance. More details in my meta-review above.

**Justification For Why Not Lower Score:**

N/A

---

### Decision · Program_Chairs · 2024-01-16

Reject